

# Cytoplasmic expression of C-MYC protein is associated with risk stratification of mantle cell lymphoma

Yi Gong[1,2], Xi Zhang[1], Rui Chen[3], Yan Wei[4], Zhongmin Zou[5] and Xinghua Chen[1]

[1] Department of Hematology, Xinqiao Hospital, The Third Military Medical University, Chongqing, China
[2] Department of Hematology-oncology, Chongqing Cancer Institute/Hospital, Chongqing, China
[3] Department of Pathology, Chongqing Cancer Institute/Hospital, Chongqing, China
[4] Department of Pathology, Xinqiao Hospital, The Third Military Medical University, Chongqing, China
[5] Institute of Toxicology, School of Preventive Medicine, The Third Military Medical University, Chongqing, China

## ABSTRACT

**Aim.** To investigate the association of C-MYC protein expression and risk stratification in mantle cell lymphoma (MCL), and to evaluate the utility of C-MYC protein as a prognostic biomarker in clinical practice.

**Methods.** We conducted immunohistochemical staining of C-MYC, Programmed cell death ligand 1 (PD-L1), CD8, Ki-67, p53 and SRY (sex determining region Y) -11 (SOX11) to investigate their expression in 64 patients with MCL. The staining results and other clinical data were evaluated for their roles in risk stratification of MCL cases using ANOVA, Chi-square, and Spearman's Rank correlation coefficient analysis.

**Results.** Immunohistochemical staining in our study indicated that SOX11, Ki-67 and p53 presented nuclear positivity of tumor cells, CD8 showed membrane positivity in infiltrating T lymphocytes while PD-L1 showed membrane and cytoplasmic positivity mainly in macrophage cells and little in tumor cells. We observed positive staining of C-MYC either in the nucleus or cytoplasm or in both subcellular locations. There were significant differences in cytoplasmic C-MYC expression, Ki-67 proliferative index of tumor cells, and CD8 positive tumor infiltrating lymphocytes (CD8+TIL) among three risk groups ($P = 0.000$, $P = 0.037$ and $P = 0.020$, respectively). However, no significant differences existed in the expression of nuclear C-MYC, SOX11, p53, and PD-L1 in MCL patients with low-, intermediate-, and high risks. In addition, patient age and serum LDH level were also significantly different among 3 groups of patients ($P = 0.006$ and $P = 0.000$, respectively). Spearman's rank correlation coefficient analysis indicated that cytoplasmic C-MYC expression, Ki-67 index, age, WBC, as well as LDH level had significantly positive correlations with risk stratification ($P = 0.000$, 0.015, 0.000, 0.029 and 0.000, respectively), while CD8+TIL in tumor microenvironment negatively correlated with risk stratification of patients ($P = 0.006$). Patients with increased positive cytoplasmic expression of C-MYC protein and decreased CD8+TIL appeared to be associated with a poor response to chemotherapy, but the correlation was not statistically significant.

**Conclusion.** Our study suggested that assessment of cytoplasmic C-MYC overexpression and cytotoxic T lymphocytes (CTLs) by immunohistochemical staining might

Corresponding authors
Zhongmin Zou,
zouzhmin@yahoo.com
Xinghua Chen, chenxh96@163.com

be helpful for MCL risk stratification and outcome prediction. However, large cohort studies of MCL patients with complete follow up are needed to validate our speculation.

## INTRODUCTION

Mantle cell lymphoma (MCL) is a less frequent subtype of B cell non-Hodgkin lymphoma characterized by t (11; 14) chromosome translocation and aggressive clinical behavior (*Smedby & Hjalgrim, 2011*). Despite new advances of therapeutic methods in recent years, MCL remains an incurable disease with poor prognosis and most patients eventually succumb to relapse after initial therapy (*Cheah, Seymour & Wang, 2016*). CHOP (cyclophosphamide, doxorubicin, vincristine, and prednisone)-like chemotherapy with or without rituximab or autologous stem cell transplantation (ASCT) is the most popular regimen applied for the treatment of MCL, with complete remission rates of 20%–50% and a median overall survival of about 3 years (*Dreyling, 2014*; *Vose, 2015*). Dose-intensified chemotherapy such as Hyper-CVAD (hyperfractionated cyclophosphamide, vincristine, doxorubicin, and dexamethasone alternating with high-dose methotrexate and cytarabine) with rituximab has achieved higher complete remission rates and 3-year failure-free survival in several clinical studies, however, it has also induced significant toxicity, which made it ineligible for most elderly and frail patients (*Khouri et al., 2016*; *Romaguera et al., 2005*). In recent years, with the advance in molecular pathogenesis of MCL, novel small molecular drugs including Bortezomib, Lenalidomide, ibrutinib, and idelalisib have been introduced into treatment of relapsed/refractory MCL and shown promising clinical outcome for part of these patients (*Inamdar et al., 2016b*). Due to genetic, pathological and clinical heterogeneity of MCL (*Inamdar et al., 2016a*), there is a critical need for reproducible biomarkers to guide the decision of individualized risk-adapted treatment for MCL patients.

C-MYC is one of the most frequently deregulated oncogenes in human cancer, and C-MYC encoded protein functions as an important transcription factor involved in the regulation of cell growth and cell cycle progression (*Dang et al., 2006*; *Sears, 2004*). In malignant cells, genetic alteration on C-MYC gene leads to consistent overexpression of C-MYC protein and promotes tumor progression (*Meyer & Penn, 2008*; *Ott, 2014*). High expression of C-MYC was reported to be associated with poor outcome of a large quantity of malignant diseases including aggressive lymphomas (*Sewastianik et al., 2014*). C-MYC abnormality has also been found in MCL patients with worse prognosis (*Choe et al., 2016*; *Hu et al., 2017*). Recently, the immune-modulatory function of C-MYC was identified in tumorigenesis (*Casey, Baylot & Felsher, 2017*; *Casey et al., 2016*), which has drawn much attention on the role of oncogenes in tumor microenvironment.

Mantle cell lymphoma international prognostic index (MIPI) has been widely used for risk stratification of advanced MCL patients. It combines clinical and laboratory

parameters to divide patients into low-, intermediate-, and high-risk groups (*Hoster et al., 2008*; *Hoster et al., 2014*). However, the correlation between C-MYC expression and MIPI, and the clinical value of C-MYC expression in treatment decision for MCL patients are not clear. In this study, we investigated the potential of C-MYC protein level assessed by immunohistochemistry for MCL risk stratification and evaluated its role in individualized therapies for patients.

## MATERIALS AND METHODS

### Case selection

Cases of MCL diagnosed between 2013 and 2016 in Chongqing Cancer Institute/Hospital were reviewed for complete information including pathological tissues and in-patient history. Sixty-four cases were identified and included in this study. All cases were reviewed by two experienced pathologists according to the criteria of the World Health Organization Classification. Data on clinical and laboratory findings such as staging, white blood cell (WBC) count, renal and liver function tests including serum albumin, globulin and β2-microglobulin (β2M), bone marrow biopsy, and imaging examinations (ultrasonic examination and radiologic examination of brain, chest, abdomen and pelvis) at the time of diagnosis were reviewed. Patients were staged according to Ann Arbor classification, and risk stratification was performed based on MIPI score system. Evaluation of therapy response was based on the Lugano classification for response assessment of Hodgkin and non-Hodgkin lymphoma (*Cheson et al., 2014*). All the cases were divided into low-, intermediate-, and high-risk groups according to MIPI score, as well as complete remission (CR), partial remission (PR) and progressive disease (PD) groups according to the therapy response. The study was approved by the ethics committees of Chongqing Cancer Institute/Hospital (No. 2017-016).

### Immunohistochemistry analysis

Immunohistochemical staining was performed using formalin-fixed paraffin-embedded tissue sections from pre-therapeutic samples according to the manufacturer's instructions. Rabbit anti-human C-MYC monoclonal antibody (clone Y69, diluted in 1:150), mouse anti-human Ki-67 monoclonal antibody (clone MIB-1, diluted in 1:200), and rabbit anti-human PD-L1 monoclonal antibody (clone SP142, diluted in 1:150) were purchased from ORIGENE (USA). Rabbit anti-human CD8 monoclonal antibody (clone SP16, diluted in 1:200), mouse anti-human SOX11 monoclonal antibody (clone MRQ-58, diluted in 1:150), and rabbit anti-human p53 monoclonal antibody (clone SP5, diluted in 1:200) were purchased from Abcam (USA). GTVision III detection system was purchased from DAKO (USA). Positivity of membranous and cytoplasmic staining pattern was scored by the staining density, ranging from 0 to 3 (0 = no staining/−, 1 = weak staining/+, 2 = moderate staining/++, 3 = strong staining/+++). Positivity of nuclear staining pattern was quantified as the percentage of positive MCL cells by manual inspection of stained sections. Tumor-infiltrating CD8 positive T cells were quantified as total counts of CD8 positive lymphocytes per high power field (HPF; 0.2 mm$^2$) by manual inspection of

stained sections with at least 10 fields of strong staining density. All the immuno-staining sections were independently determined by 2 experienced pathologists in a blinded fashion.

## Statistical analysis

All data were analyzed with SPSS 18.0 (IBM Corporation, NY, USA). Categorical variables were compared using chi-square test. The difference between continuous variables was assessed using ANOVA. The role of above clinicopathological parameters for MCL risk stratification and therapy individualization were evaluated based on MIPI score and clinical response respectively. $P$ value of less than 0.05 was considered statistically significant. Spearman's rank correlation coefficient analysis was performed to assess the association between clinical factors and immunohistochemical staining results of C-MYC, Ki-67, PD-L1, CD8, SOX11 and p53.

## RESULTS

### Patient characteristics

There were 51 male (79.7%) and 13 female (20.3%) patients. The median age was 60 years old (range 21–87 years) at the time of diagnosis. The majority of the cases (71.9%) presented as nodal diseases, and 18 cases occurred in primary extra-nodal sites including colon, rectum, ileocecum, oropharynx, and spleen. Sixty-one (95.3%) patients were in Ann Arbor advanced stage (III–IV) and 35 (54.7%) patients presented with B symptoms. Only 4 (6.4%) patients had an ECOG performance status (PS) of 2–4. Thirty-one of the 64 patients had treatment information of at least 2 cycles of chemotherapy for response evaluation. As depicted in Table S1, 13 patients received CHOP, 11 patients received rituximab combined with CHOP, 3 patients received CHOPE, 3 patients received Hyper-CVAD and 1 patient received GDP treatment. After 2 treatment cycles, 5 patients achieved CR, 17 patients achieved PR, and 9 patients demonstrated PD.

### Immunohistochemical study

As shown in Fig. 1, SOX11, Ki-67 and p53 presented nuclear positivity of tumor cells, and CD8 showed membrane positivity of T lymphocytes infiltrating the microenvironment. PD-L1 showed membrane and cytoplasm positive pattern mainly in macrophage cells and little in tumor cells. However, C-MYC staining was observed either in the nucleus or cytoplasm or in both subcellular locations.

As shown in Table 1, there were significant differences in cytoplasmic C-MYC expression, Ki-67 proliferative index of tumor cells, and CD8 positive tumor infiltrating lymphocytes (CD8+TIL) among three risk groups ($P = 0.000$, $P = 0.037$, $P = 0.020$). However, no significant difference existed in the expression of nuclear C-MYC, SOX11, p53, and PD-L1 among MCL patients with low, intermediate, and high risks, respectively. In addition, other clinical parameters including age and LDH level showed significant difference among 3 groups of patients as defined by MIPI score ($P = 0.006$, $P = 0.000$).

Spearman's rank correlation coefficient analysis was performed to evaluate the relationship between clinicopathologcial parameters and prognosis or therapy response. As shown in Table 1, patients' age, WBC, LDH level, as well as cytoplasmic C-MYC

**Table 1   Clinical and immunohistochemical staining characteristics of patients grouped by risk.**

| Variable | n | Low risk | Intermediate risk | High risk | P value |
|---|---|---|---|---|---|
| **Sex** | 64 | | | | .117 |
|     Male | 51 (79.7%) | 15 (23.4%) | 27 (42.2%) | 9 (14.1%) | |
|     Female | 13 (20.3%) | 5 (7.8%) | 3 (4.7%) | 5 (7.8%) | |
| Age (years) | 64 | $54.10 \pm 8.60$ | $61.77 \pm 7.77$ | $64.86 \pm 15.39$ | .006[**] |
| WBC ($10^9$/L) | 64 | $6.50 \pm 2.17$ | $8.07 \pm 3.69$ | $8.93 \pm 3.19$ | .078 |
| LDH (U/L) | 64 | $190.03 \pm 74.84$ | $341.35 \pm 226.03$ | $521.15 \pm 266.44$ | .000[**] |
| **ECOG performance status** | 64 | | | | 0.112 |
|     **0** | 46 (71.9%) | 16 (25.0%) | 22 (34.4%) | 8 (12.5%) | |
|     **1** | 14 (21.9%) | 4 (6.3%) | 7 (10.9%) | 3 (4.7%) | |
|     **2** | 4 (6.2%) | 0 (0.0%) | 1 (1.6%) | 3 (4.7%) | |
| **Ann Arbor stage** | 64 | | | | .302 |
|     II | 3 (4.69%) | 2 (3.1%) | 1 (1.6%) | 0 (0.0%) | |
|     III | 32 (50.0%) | 11 (17.2%) | 12 (18.8%) | 9 (14.1%) | |
|     IV | 29 (45.31%) | 7 (10.9%) | 17 (26.6%) | 5 (7.8%) | |
| **Location** | 64 | | | | 0.589 |
|     Nodal | 46 (71.9%) | 16 (25.0%) | 20 (31.3%) | 10 (15.6%) | |
|     Extra-nodal | 18 (28.1%) | 4 (6.3%) | 10 (15.6%) | 4 (6.3%) | |
| **B symptoms** | 64 | | | | .363 |
|     No | 29 (45.3%) | 10 (15.6%) | 15 (23.4%) | 4 (6.3%) | |
|     Yes | 35 (54.7%) | 10 (15.6%) | 15 (23.4%) | 10 (15.6%) | |
| **Response** | 31 | | | | .883 |
|     CR | 5 (7.8%) | 2 (6.5%) | 3 (9.7%) | 0 (0.0%) | |
|     PR | 17 (26.56%) | 6 (19.4%) | 8 (25.8%) | 3 (9.7%) | |
|     PD | 9 (14.1%) | 3 (9.7%) | 5 (16.1%) | 1 (3.2%) | |
| **Cytoplasmic C-MYC** | 64 | | | | .000[**] |
|     − | 19 (29.7%) | 11 (17.2%) | 7 (10.9%) | 1 (1.6%) | |
|     + | 25 (39.1%) | 6 (9.4%) | 17 (26.6%) | 2 (3.1%) | |
|     ++ | 20 (31.2%) | 3 (4.7%) | 6 (9.4%) | 11 (17.2%) | |
| Nuclear C-MYC (%) | 64 | $16 \pm 20$ | $11 \pm 13$ | $16 \pm 19$ | .537 |
| p53 (%) | 61 | $19 \pm 21$ | $13 \pm 17$ | $30 \pm 31$ | .082 |
| CD8+TIL(/0.2 mm²) | 61 | $160.00 \pm 55.52$ | $132.07 \pm 54.60$ | $105.71 \pm 45.69$ | .020[*] |
| **PD-L1** | 48 | | | | .599 |
|     − | 15 (31.3%) | 6 (12.5%) | 6 (12.5%) | 3 (6.3%) | |
|     + | 28 (58.3%) | 4 (8.3%) | 17 (35.4%) | 7 (14.6%) | |
|     ++ | 4 (8.3%) | 1 (2.1%) | 2 (4.2%) | 1 (2.1%) | |
|     +++ | 1 (2.1%) | 0 (0.0%) | 1 (2.1%) | 0 (0.0%) | |
| **SOX11 (%)** | 64 | $33 \pm 31$ | $29 \pm 31$ | $29 \pm 27$ | .887 |
| **Ki67 (%)** | 64 | $31 \pm 16$ | $45 \pm 25$ | $50 \pm 26$ | .037[*] |

**Notes.**
[*]$P < 0.05$.
[**]$P < 0.01$.

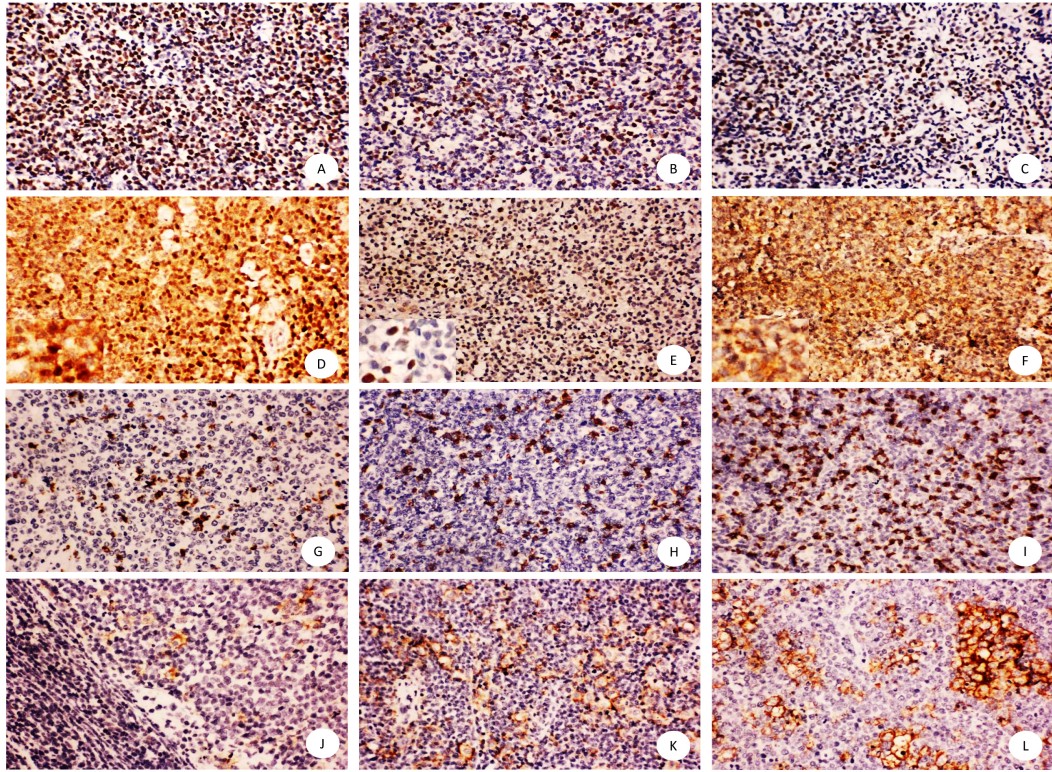

**Figure 1** **Immunohistochemical staining of mantle cell lymphoma using anti-SOX11, Ki67, p53, C-MYC, CD8, PD-L1.** Shown are representative staining patterns of SOX11 (A), Ki-67 (B), p53 (C), C-MYC (D–F), CD8 (G–I), PD-L1 (J–L). Original magnification, ×200. Inserts: typical cytoplasmic and nuclear staining of C-MYC (D), nuclear staining of C-MYC (E), and cytoplasmic staining of C-MYC (F), Original magnification, ×400.

expression and Ki-67 index demonstrated a significantly positive correlation with risk stratification ($P = 0.000$, 0.029, 0.000, and 0.000, respectively), while CD8+TIL in tumor microenvironment negatively correlated with risk stratification of patients ($P = 0.006$). Increased positive cytoplasmic expression of C-MYC protein and decreased CD8+TIL were associated with a poor response to chemotherapy, but the correlation did not reach statistical significance (Tables 2 and 3).

## DISCUSSION

Assessment of C-MYC oncogene condition is critical for differentiating diagnosis and predicting prognosis in Burkitt lymphoma and diffuse large B cell lymphoma harboring a C-MYC translocation (*Dalla-Favera et al., 1982*; *Ott, Rosenwald & Campo, 2013*; *Savage et al., 2009*). As a potent nuclear transcription factor, C-MYC protein overexpression has been typically found in the nucleus of lymphoma cells (*Choe et al., 2016*; *Oberley et al., 2013*). However, in this study, we observed three patterns of C-MYC expression including nuclear, cytoplasmic, and both nuclear and cytoplasmic localization in mantle cell lymphoma (Fig. 1), which was different from the results reported by Matthew J. et al. using commercially available C-MYC monoclonal antibody (clone number: Y69). The immunohistochemistry

**Table 2** Clinical and immunohistochemical staining characteristics of patients grouped by treatment response.

| Variable | n | CR | PR | PD | P value |
|---|---|---|---|---|---|
| **Sex** | 31 | | | | .088 |
| Male | 23 (74.2%) | 5 (16.1%) | 10 (32.3%) | 8 (25.8%) | |
| Female | 8 (25.8%) | 0 (0.0%) | 7 (22.6%) | 1 (3.2%) | |
| Age (years) | 31 | 52.60 ± 3.91 | 56.24 ± 12.73 | 56.89 ± 8.89 | .760 |
| WBC ($10^9$/L) | 31 | 10.06 ± 3.61 | 7.68 ± 3.78 | 7.54 ± 3.43 | .407 |
| LDH (U/L) | 31 | 334.82 ± 243.60 | 333.21 ± 269.65 | 273.31 ± 95.30 | .804 |
| **ECOG performance status** | 31 | | | | 0.130 |
| 0 | 21 (67.7%) | 4 (12.9%) | 13 (41.9%) | 4 (12.9%) | |
| 1 | 8 (25.8%) | 1 (3.2%) | 2 (6.5%) | 5 (16.1%) | |
| 2 | 2 (6.5%) | 0 (0.0%) | 2 (6.5%) | 0 (0.0%) | |
| **Ann Arbor stage** | 31 | | | | .905 |
| II | 1 (3.2%) | 0 (0.0%) | 1 (3.2%) | 0 (0.0%) | |
| III | 16 (51.6%) | 3 (9.7%) | 8 (25.8%) | 5 (16.1%) | |
| IV | 14 (45.2%) | 2 (6.5%) | 8 (25.8%) | 4 (12.9%) | |
| **Location** | 31 | | | | .437 |
| Nodal | 23 (74.2%) | 3 (9.7%) | 12 (38.7%) | 8 (25.8%) | |
| Extra-nodal | 8 (25.8%) | 2 (6.5%) | 5 (16.1%) | 1 (3.2%) | |
| **B symptoms** | 31 | | | | .844 |
| No | 15 (48.4%) | 3 (9.7%) | 8 (25.8%) | 4 (12.9%) | |
| Yes | 16 (51.6%) | 2 (6.5%) | 9 (29.0%) | 5 (16.1%) | |
| **Risk group** | 31 | | | | .720 |
| Low | 11 (35.5%) | 2 (6.5%) | 6 (19.4%) | 3 (9.7%) | |
| Intermediate | 16 (51.6%) | 3 (9.7%) | 8 (25.8%) | 5 (16.1%) | |
| High | 4 (12.9%) | 0 (0.0%) | 3 (9.7%) | 1 (3.2%) | |
| **Cytoplasmic C-MYC** | 31 | | | | .031[*] |
| − | 11 (35.5%) | 2 (6.5%) | 8 (25.8%) | 1 (3.2%) | |
| + | 11 (35.5%) | 3 (9.7%) | 2 (6.5%) | 6 (19.4%) | |
| ++ | 9 (29.0%) | 0 (0.0%) | 7 (22.6%) | 2 (6.5%) | |
| Nuclear C-MYC (%) | 31 | 17 ± 12 | 15 ± 17 | 6 ± 7 | .227 |
| p53 (%) | 31 | 16 ± 21 | 20 ± 23 | 20 ± 23 | .950 |
| CD8+TIL(/0.2 mm$^2$) | 31 | 154.00 ± 35.78 | 145.00 ± 55.74 | 111.11 ± 64.51 | .277 |
| **PD-L1** | 24 | | | | .352 |
| − | 11 (45.8%) | 1 (4.2%) | 5 (20.8%) | 5 (20.8%) | |
| + | 11 (45.8%) | 3 (12.5%) | 6 (25.0%) | 2 (8.3%) | |
| ++ | 2 (8.3%) | 0 (0.0%) | 2 (8.3%) | 0 (0.0%) | |
| **SOX11 (%)** | 31 | 36 ± 45 | 26 ± 30 | 34 ± 31 | .757 |
| **Ki67 (%)** | 31 | 38 ± 16 | 50 ± 27 | 38 ± 30 | .480 |

**Notes.**
[*]$P < 0.05$.

**Table 3** Correlation coefficient analysis for different parameters with risk group.

| Variable | n | Spearman correlation coefficient with risk group | P value |
|---|---|---|---|
| Sex | 64 | 0.055 | 0.669 |
| Age | 64 | 0.454** | 0.000 |
| WBC | 64 | 0.273* | 0.029 |
| LDH | 64 | 0.593** | 0.000 |
| ECOG performance status | 64 | 0.207 | 0.101 |
| Ann Arbor stage | 64 | 0.078 | 0.542 |
| Location | 64 | 0.085 | 0.502 |
| B symptoms | 64 | 0.138 | 0.278 |
| Response | 31 | 0.058 | 0.755 |
| Cytoplasmic C-MYC | 64 | 0.496** | 0.000 |
| Nuclear C-MYC | 64 | 0.005 | 0.968 |
| p53 | 61 | 0.079 | 0.544 |
| CD8+TIL | 61 | −0.351** | 0.006 |
| PD-L1 | 48 | 0.163 | 0.268 |
| SOX11 | 64 | −0.020 | 0.873 |
| Ki67 | 64 | 0.303* | 0.015 |

Notes.
*$P < 0.05$.
**$P < 0.01$.

staining procedures have been independently validated by two experienced pathologists to exclude false positive or non-specific results. We searched a few studies that reported cytoplasmic expression of C-MYC in leukemia cell line, endometrial carcinoma, and high grade B cell lymphomas (*Craig et al., 1993*; *Geisler et al., 2004*; *Ruzinova, Caron & Rodig, 2010*), but all the authors defined the cytoplasmic C-MYC status as negative or positive, which was different with our four-categories evaluation methods (0, +, ++, +++). The mechanism and biologic importance for the cytoplasmic overexpression of C-MYC was unclear. Since C-MYC needs to dimerize with Max to bind the E-box to activate its downstream genes in transformed cells (*Dang et al., 2006*), accumulation of C-MYC protein in the cytoplasm might suggest an unknown deregulated pathway synergized with other pathways in promoting tumor growth. Therefore, the cytoplasmic pattern of C-MYC expression might indicate a worse prognosis compared with the typical nuclear pattern. In this study, we found a positive correlation only between cytoplasmic C-MYC level and risk stratification of MCL, but not between nuclear C-MYC level and risk stratification of MCL, suggesting the potential clinical application of C-MYC immunohistochemical staining in determining prognosis and treatment of MCL patients.

PD-L1 is an important immune checkpoint molecule in tumor microenvironment, which is associated with immune evasion in a variety of malignancies (*Dong & Chen, 2003*; *Janakiram et al., 2016*; *Wlasiuk, Putowski & Giannopoulos, 2016*). Prevalence of Th1 type lymphocytes such as CD8+ T lymphocytes usually predicts good prognosis of cancer patients (*Donnem et al., 2015*; *Hanahan & Weinberg, 2011*). Expression of PD-L1 and infiltration of cytotoxic T lymphocytes (CTLs) in tumor microenvironments are

reported to be the prerequisite for effective response of PD-L1 pathway blockade therapy in many cancer patients (*Ock et al., 2016*; *Teng et al., 2015*). High level of PD-L1 expression has been observed in some types of lymphomas including a subset of aggressive B cell lymphomas and EBV-associated malignancies such as classical Hodgkin lymphoma (CHL) (*Berghoff et al., 2014*; *Chen et al., 2013*; *Kwon et al., 2016*). In our study, positive PD-L1 staining was found in 68.75% (33/48) MCL patients but only 1.04% (5/48) cases were recorded as moderate to strong positivity. In addition, PD-L1 positive cells were mainly macrophages in the microenvironment. There was no significant difference in PD-L1 expression among various risk groups, but the density of CD8+ T lymphocytes showed a negative correlation with risk stratification, suggesting that CD8+ T lymphocytes might be a useful prognostic biomarker for risk stratification of MCL patients, consistent with other previous studies (*Nygren et al., 2014*). Extensive future study of PD1/PD-L1 pathways in a large cohort of MCL patients is also warranted.

Ki-67 as an independent, significant prognostic factor for MCL has been proved in many clinical studies and integrated into MIPI score system as a combined biological index (MIPI$_b$) (*Hoster et al., 2008*). The significance of Ki-67 was also confirmed in our study. We similarly assessed the expression of another two important biomarkers, SOX11 (*Ek et al., 2008*; *Xu & Li, 2010*) and p53 (*Nordstrom et al., 2014*; *Tessoulin et al., 2017*), in MCL cases but found no significant difference among low, intermediate, and high risk group of MCL patients.

We also noted that patients with increased positive cytoplasmic expression of C-MYC protein and decreased CD8+TIL were associated with poor response to chemotherapy, but the correlation did not reach statistical significance, probably due to the loss of follow-up data in 33 of 64 patients in our cohort.

Limitations of the study included small cohort of cases and insufficient survival data because of the difficulties in follow-up for most lymphoma patients of this area. In addition, the systematic bias including tissue fixation and observation might influence the accuracy of assessment for protein biomarkers in immunohistochemistry studies (*Torlakovic et al., 2010*). Nevertheless, this is the first report on the association of aberrant expression of C-MYC protein, CD8+TIL, and Ki-67 with risk stratification of MCL patients.

In conclusion, our work suggested that assessment of cytoplasmic C-MYC overexpression, CD8 positive CTLs, and Ki-67 by immunohistochemical staining might be helpful for risk stratification and prognosis of MCL patients. Large cohort studies of MCL patients with complete follow up are needed to further examine the potential of these biomarkers being integrated into routine pathological work.

### Funding

This study was supported by the National Natural Science Foundation of China (81170467 and 81270569), Major Project of PLA Medical S&T foundation (AWS11C004) and Medical Science Research Foundation of Chongqing Health and Family Planning Committee

(2015MSXM224). The funders had no role in study design, data collection and analysis, decision to publish, or preparation of the manuscript.

## Grant Disclosures

The following grant information was disclosed by the authors:

National Natural Science Foundation of China: 81170467, 81270569.

Major Project of PLA Medical S&T foundation: AWS11C004.

Medical Science Research Foundation of Chongqing Health and Family Planning Committee: 2015MSXM224.

## Competing Interests

The authors declare there are no competing interests.

## Author Contributions

- Yi Gong and Rui Chen conceived and designed the experiments, performed the experiments, analyzed the data, contributed reagents/materials/analysis tools, wrote the paper, prepared figures and/or tables.
- Xi Zhang and Xinghua Chen conceived and designed the experiments, reviewed drafts of the paper.
- Yan Wei performed the experiments, analyzed the data, contributed reagents/materials/analysis tools.
- Zhongmin Zou reviewed drafts of the paper.

## Human Ethics

The following information was supplied relating to ethical approvals (i.e., approving body and any reference numbers):

The study was approved by the ethics committees of Chongqing Cancer Hospital/ Chongqing Cancer Institute.

## Supplemental Information

Supplemental information for this article can be found online at http://dx.doi.org/10.7717/ peerj.3457#supplemental-information.

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
