# Peer review of "Cytoplasmic expression of C-MYC protein is associated with risk stratification of mantle cell lymphoma"

_PeerJ, doi:10.7717/peerj.3457_

## Round 0.1 · original submission · Major Revisions

I agree with comments and criticisms of reviewers. Please address their comments. The main ones are 1) Don't be definitive that prognostic markers should be standard without ANY data that you know how to change treatment based on this data. 2) Show some humility. Hundreds of prognostic markers have been found in lymphomas. P values don't necessarily change human clinical outcomes. 3) Grammar. 4) Correlate MYC testing by different methods and other perspectives (references) in the field.

Reviewer 1 ·

Basic reporting

The manuscript readswell with clear language. Construction of the sentences could be better though- For example, the abstract reads mostly in "third person". In line 14 and onwards, you the authors could mention "we conducted immunohistochemical staining for..." or "we observed c-MYC either in nucleus...". References needs to be mentioned when discussing MCL therapy. For example, no mention of Hyper-CVAD regimen and ibrutinib for relapsed-refractory patients. The conclusions, in my opinion, went a bit far when recommended routine staining of c-MYC. I think the study showed the value of such staining for risk-stratification but needs a more robust validation.

Experimental design

The experimental design could be better described. For example, description of each of the groups being compared and the reason behind comparing needs to be reported. I recommend identifying upfront what are the groups: based on MIPI, based on clinical response and then state how are they being compared. With all fairness, P-values should be less emphasized.

Validity of the findings

Again, you should recommend that there is value of c-MYC staining and future work should focus on validation in a larger cohort before recommending routine testing. How would that impact the cost? Does this justify that? If yes, please mention why.

Additional comments

A nice work looking at clinical implication of c-MYC staining along with others. The work needs some revisions, especially re: intro when you are discussing MCL treatment options currently. I think if you do not mention hyper-CVAD alongside CHOP and talk about ibrutinib- this is not giving the clear picture.

Reviewer 2 ·

Basic reporting

The manuscript is clearly written and abide to the structure of PeerJ

Experimental design

IHC staining method was utilized to measure the expression of C-MYC, PD-L1, CD8, Ki-67, p53, and SOX-11 in 64 patients with MCL. The methods and design are reasonable to answer their scientific questions.

Validity of the findings

no comment

Additional comments

In this manuscript the authors propose to investigate the association between myc protein expression and risk stratification in MCL, and to evaluate the utility of myc protein as a prognostic biomarker for MCL. IHC staining method was utilized to measure the expression of C-MYC, PD-L1, CD8, Ki-67, p53, and SOX-11 in 64 patients with MCL. Their findings indicate that while PD-L1 expression, mainly in the tumor microenvironment, did not show significant difference among various risk groups, but the density of CD8+ T lymphocytes, on the other hand, showed a negative correlation with risk stratification. The expression of p53 and SOX-11 also showed no significant correlations. Of interest is that they showed that cytoplasmic expression of c-myc protein and decreased CD8+ TIL were associated with poor response to chemotherapy, but the correlation did not reach statistical significance. The finding that cytoplasmic MYC level correlates with risk stratification is important and could be clinically relevant. However, larger cohort studies of MCL patients is required to validate this finding. The quality of Figure 1 needs attention. I would recommend to provisionally accept this manuscript for publication with the following comments:

1. Figure 1, images are too blurry, suggest enhancing the resolution or show higher magnification.
2. Title could be misleading.
3. Does cytoplasmic MYC protein expression correlates with myc gene abnormalities (gene amplification/translocation)?

Reviewer 3 ·

Basic reporting

The authors use language that is clear and concise to present their manuscript. There are some minor grammatical and potentially typographic errors that need to be addressed (see below).

The introduction is well-written and provides context to their presented work. The manuscript is structured with appropriate subheadings, figures are easy to follow and raw data is provided as required by PeerJ.

Experimental design

The authors present their original research which shows interesting findings that may be clinically relevant and may add to the risk stratification of mantle cell lymphoma. The number of patients that have follow-up is small and survival data is not available which they acknowledge as a limitation in their paper.

The methods section is detailed and includes the names and sources of the antibodies/machines used allowing for replication of their experiment by readers. Although the scoring of cytoplasmic MYC staining is subjective, this was addressed in the methods by independent blinded review by 2 experienced pathologists. They used different scoring systems for measuring cytoplasmic and nuclear staining patterns. It will be helpful to understand why different quantification systems were used?

Most published data in lymphomas and other malignancies correlates nuclear MYC staining with MYC amplification and prognosis. Did the authors consider looking at the correlation between nuclear and cytoplasmic MYC staining? Could nuclear or cytoplasmic or both have different prognostic implications? If there were limitations in looking at this, they should be acknowledged or the authors reasoning explained.

Validity of the findings

The authors report a single institution study of 64 patients with MCL and show a correlation between cytoplasmic MYC staining on immunohistochemistry with risk stratification by the MIPI score. They also report a negative association of CD8+ tumor infiltrating lymphocytes with risk stratification.

The clinical data (available only 31 patients) is incomplete as only interim disease response after 2 cycles of chemotherapy is available. Longer follow-up with response at the end of first line treatment or survival data would be more meaningful.

The authors should consider including the following papers that are relevant to their discussion and present different findings from their own: Oberley et al. Immunohistochemical evaluation of MYC expression in mantle cell lymphoma. Histopathology. 2013 Oct.

Ref 21, shows nuclear C-MYC staining to have worse outcomes than cytoplasmic staining in endometrial cancer. Consider further discussion on why the authors report different results.

Additional comments

- Lines 68-69: Please spell out the abbreviations used.
- Line 85, sentence starting with “And”
- In line 99, recommend including median age with range in place of mean.
- In line 103, the authors state 32 of 64 patients had treatment information but outcome information is reported on only 31 patients. This should be stated more clearly.
- Line 118-120: the authors state that age and LDH level were significantly different among the 3 risk groups. In the methods, the risk groups have been defined by using MIPI which includes age, LDH, WBC and ECOG performance status as factors defining prognosis and hence they are expected to be different among the low, intermediate and high risk groups that the authors have compared. This is not a finding but a result of the method used and should be stated as such.
- Line 121, “furtherly” grammatical error.
- Lines 122, the authors likely mean Table 1 or 3 instead of table 2. Please correct.
- TABLES: The tables can be confusing for the reader and need to be more clear regarding what is the denominator used when reporting percentages.
o Consider including total numbers under the column headings.
o Please include units of reporting for variables such as age, WBC, LDH and immunohistochemistry.
o The percentages in Table 1 under PDL1 by risk group do not add up.

---

## Round 0.2 · accepted · Accept

Thank you for reviewing and making revisions as suggested by the reviewers.